# Newborn Screening—A Worldwide Endeavour to Protect

**DOI:** 10.3390/ijns11030080

**Published:** 2025-09-18

**Authors:** James R. Bonham, Dianne Webster, Amy Gaviglio, Aysha Habib Khan, R. Rodney Howell, Peter C. J. I. Schielen

**Affiliations:** 1International Society for Neonatal Screening, Reigerskamp 273, 3607 HP Maarssen, The Netherlands; j.bonham@nhs.net; 2National Newborn Screening Laboratory, LabPlus, Health New Zealand Te Whatu Ora, Auckland 1023, New Zealand; diannew@adhb.govt.nz; 3Connetics Consulting, LLC, Minneapolis, MN 55417, USA; amy.gaviglio@outlook.com; 4Department of Pathology and Lab Medicine, Aga Khan University, Karachi 74800, Pakistan; aysha.habib@aku.edu; 5Department of Medicine, Aga Khan University, Karachi 74800, Pakistan; 6Hussman Institute for Human Genomics, Miller School of Medicine, University of Miami, Miami, FL 33136, USA; rhowell@med.miami.edu

**Keywords:** global collaboration, resilience, global challenges, instability, neonatal screening newborn screening

## Abstract

For more than 60 years, newborn (or neonatal) screening has flourished through global collaboration, demonstrating that collective action is key to success. This unity proved to be especially vital during the COVID-19 pandemic, when, despite severe disruptions, NBS services were largely preserved, reflecting the high value placed on early detection and care for vulnerable newborns. Today, the International Society for Neonatal Screening (ISNS) recognises that NBS programmes face increasing challenges due to global instability. While direct assistance is not always possible, ISNS emphasises the strength of the international NBS community—scientists, clinicians, patient groups, and industry partners—who are committed to mutual support and knowledge-sharing. Building on the proud legacy inspired by pioneers like Bob Guthrie, this community is enriched by diverse voices and is unified by a shared vision: to ensure that all children with rare disorders have access to life-saving screening and care. Safeguarding and advancing this foundation is a responsibility owed to future generations.

There are few things that are of greater value than initiatives that reduce morbidity and mortality by effective interventions made shortly after birth. Such investment made on behalf of the coming generation speaks well of society and can offer hope and a future to families facing the most challenging of circumstances. It is, therefore, unsurprising that many have described newborn (or neonatal) screening (NBS) as “one of the most successful public health interventions of the twentieth century” [1].

The international exchange of ideas, technology, and organisational planning were the foundation of these life-changing initiatives. When, during 1964–1967, Bob Guthrie, working in the University of Buffalo, New York, developed assays to detect galactosaemia, maple syrup urine disease, phenylketonuria, homocystinuria, and tyrosinaemia using dried blood spot samples, his immediate instinct was to share his insights and vision with others around the world. He travelled from the USA from New Zealand, and, on his return, he set out with his family in a new VW campervan to travel to Australia, South Africa, Spain, Italy, Switzerland, Germany, Scandinavia, and the British Isles. While he often carried with him crucial pieces of technology, such as dried blood spot punches, he also carried an even more important asset: an international vision and an ability to inspire while listening to others.

Indeed, the effectiveness of Guthrie’s new screening approach could only gain maximum effect by harnessing the developments in treatment, such as an effective dietary treatment for phenylketonuria, made a decade earlier by pioneers in Europe such as Louis Woolf and Horst Bickel.

From the beginning, newborn screening was the result of effective international collaboration, and we share a collective duty to follow that fruitful tradition. We can celebrate the fact that, today, approximately 40 million babies around the world are offered newborn screening, leading to the opportunity for early recognition and effective treatment for almost 40,000 babies per year. We must, however, not rest upon the vision and achievements of others, but shoulder the responsibility of continuing to develop the potential of newborn screening not just for those who already enjoy its benefits, but also for the 100 million babies born each year for whom no route to early detection and treatment currently exists.

Recently, it has been immensely encouraging to see that, during 2024 and 2025, the World Health Organisation proposed important resolutions designed to emphasise the benefits of adopting newborn screening in its widest sense in all healthcare settings [2,3]. The anticipated reduction in both mortality and morbidity, along with potential positive cost benefits, could provide important gains, even for low- and middle-income countries. The WHO are conscious that only by working together in initiatives that support sustainable development can this be achieved, and they are currently widely consulting within an international forum to help establish a framework to achieve that goal.

The International Society for Neonatal Screening held its first meeting under that name in 1991 and currently supports over 500 members in more than 80 countries worldwide. Its membership is widely drawn from laboratory scientists, physicians, genetic counsellors, nurses, economists, and ethicists, and represents those working in many cultural settings. The ISNS is committed to the original vision of providing well-organised and appropriate newborn screening, which can lead to early recognition and effective treatment for the baby concerned. At the same time, the ISNS supports training the next generation of health professionals in genomics, metabolic medicine, and laboratory diagnostics by facilitating global collaboration and prioritising knowledge-sharing and mentorship to build resilient human resource capacities across all regions.

We recognise that newborn screening is a programme and not just a test. Indeed, there are many examples that convincingly demonstrate that only with carefully co-ordinated planning emphasising the importance of the timely delivery of a baby into appropriate treatment can we gain the maximum impact. However, the journey does not end with the initiation of treatment. Sustainable NBS programmes must ensure structured, long-term follow-up, including care coordination, transition to adult services, and support for families. Only through this continuum of care can we fully realise the promise of early detection.

This requires a team effort and a safe environment which allows for and encourages the free flow of ideas, technology, and goods and services. Indeed, it is not just the scientists, physicians, and other health professionals involved who allow this activity to flourish and grow, but also our industry colleagues, who innovate and provide technical solutions and treatments, and governments, who facilitate the free exchange of knowledge and such technical solutions. The ISNS recently recognised the invaluable role of industry with the creation of a sustaining member programme allowing for industry to become an integral part of the team so that both spheres can grow together.

In recent years, newborn screening has gained unprecedented global momentum, driven by the collaborative efforts of key international bodies such as the ISNS-IFCC Global Task Force on Newborn Screening, the International Coordination Committee by the ISNS, and the World Health Organisation. These initiatives have converged to foster dialogue, support strategic frameworks, and advocate for the integration of NBS into national health systems, especially in LMICs. The WHO’s recent recommendations and draft resolutions signal a major policy shift, recognising newborn screening as not merely a technological intervention, but also as a core component of child health and universal health coverage. When translated into national policies, these global commitments have the power to catalyse funding, infrastructure development, and political will, which are essential for the sustainable implementation of newborn screening in resource-constrained settings to close the equity gap and ensure that every child, regardless of geography, has access to early detection and life-saving treatment.

Governmental agencies have a key role to play in this process, as they help support and develop newborn screening in their own countries while sharing ideas, information, and support to the wider community. The Centers for Disease Control and Prevention (CDC) and the Health Resources and Services Administration (HRSA) provide a shining example of the benefit to national policy provided by a worldwide vision. Indeed, CDC’s Newborn Screening and Molecular Biology Branch (NSMBB), while helping state newborn screening programmes creates reference materials, improves and creates new methods, supports laboratory quality, and offers technical support. These resources help the early detection of rare but serious conditions in newborns, leading to better health outcomes with reliable screening practices in various settings.

The wider community is also important. This community includes the identified babies and their families, disorder-specific advocacy groups, and the wider rare-disorder advocacy organisations. Without the collaboration of the community, programmes may not be expanded, appropriate therapies may not be funded, and outcome (long-term follow-up) studies may not be realised.

This brings us to the challenges that we face today as we seek to maintain and improve the national programmes that are already in place and extend them to others.

We live in a world of unrest in the Middle East, Europe, South China seas, parts of Africa, and other regions. This unrest brings organisational and financial instability for many. Despite these challenges, the free exchange of ideas within the wider international community is vital as we seek to unlock the potential of new technologies to detect and treat those in need. Similarly, a fair and sustainable international economic environment is paramount to enable our industry colleagues working in this sector to secure the investment needed to stimulate research and development that leads to new analytical solutions and treatments.

We therefore make a plea to policymakers and governments to recognise the unique value of newborn screening to offer the best life chances for vulnerable children. More than 60 years of experience drawn from colleagues around the world tells us that we can do this best if we do it together.

We have navigated disruption on a global scale during the recent COVID-19 pandemic, with several reports indicating that services were preserved and maintained despite severe challenges in the healthcare sector, and this offers welcome testimony to the high regard in which newborn screening is held.

The ISNS is aware that, in current times, NBS programmes are struggling with global instability in various ways. While we cannot promise direct help, we are aware of the difficulties that some are facing, and we can offer the benefits of an international community of scientists, physicians, patient-support groups, and industry colleagues committed to supporting one another where possible.

We have a proud legacy upon which to build, and we owe it to future generations to protect the things that we value. Just as Bob Guthrie inspired a generation of scientists, today’s programmes are enriched by the shared experiences and the international voices of parents and patient advocates, scientists, physicians, industry, and governments working together. The newborn screening community shares the vision of benefiting children and families with rare disorders around the world. We must protect and develop this valuable foundation for the future.

## Data Availability

No new data were created for this opinion article.

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
