# Peer review of "Newborn Screening—A Worldwide Endeavour to Protect"

_2409-515X, 2025, doi:10.3390/ijns11030080_

Round 1
Reviewer 1 Report
Comments and Suggestions for Authors
The newborn screening program and its history is very well described. It is not enough to screen (make a test), we must implement all the facets of this program, which goes from sampling to taking care of affected children.
The text therefore quite rightly mentions the stakeholders for the programme to be carried out. It also describes that it must evolve in parallel with the new technologies and treatments available. However, acceptance of the newborn screening program, and therefore families' knowledge of its benefits, is very important. The "patient support groups" and the "international voice of parents and patient advocates" are only mentioned at the end of the text. Mentioning them earlier, or even in a dedicated paragraph, should be considered. They must be the relay for the benefits of newborn screening, access to innovative treatments, and inclusion in national and international registries to participate in research in this field. Privacy regulations will not be a barrier to testing, inclusion of patients in a registry, biobanking of samples, …, if families understand the benefits of the newborn screening program and are aware of the need to evolve it.
At a time when countries' finances are geared towards other priorities and the newborn screening program includes rare diseases, it is often difficult to make equity heard (some countries screen for 1 disease, others 49). Briefly addressing the cost-benefit and not just the reduction in mortality and morbidity rates is also a point taht could be considered.
Author Response
Reviewer 1
We thank reviewer 1 for the thoughtful review and support of newborn screening programmes.
The newborn screening program and its history is very well described. It is not enough to screen (make a test), we must implement all the facets of this program, which goes from sampling to taking care of affected children.
The text therefore quite rightly mentions the stakeholders for the programme to be carried out. It also describes that it must evolve in parallel with the new technologies and treatments available. However, acceptance of the newborn screening program, and therefore families' knowledge of its benefits, is very important. The "patient support groups" and the "international voice of parents and patient advocates" are only mentioned at the end of the text. Mentioning them earlier, or even in a dedicated paragraph, should be considered. They must be the relay for the benefits of newborn screening, access to innovative treatments, and inclusion in national and international registries to participate in research in this field. Privacy regulations will not be a barrier to testing, inclusion of patients in a registry, biobanking of samples, …, if families understand the benefits of the newborn screening program and are aware of the need to evolve it.
This is an important point and the following paragraph has been added “ The wider community too are important. They are the identified babies and their families, the disorder specific advocacy groups and the wider Rare Disorder advocacy organizations. Without the collaboration of the community programmes may not be expanded, appropriate therapies funded or outcome (long term followup) studies realized.”
At a time when countries' finances are geared towards other priorities and the newborn screening program includes rare diseases, it is often difficult to make equity heard (some countries screen for 1 disease, others 49). Briefly addressing the cost-benefit and not just the reduction in mortality and morbidity rates is also a point taht could be considered. We agree and have modified the sentence “The anticipated reduction in both mortality and morbidity, along with potential positive cost benefits, could be important gains, even for low and middle income countries.”
Reviewer 2 Report
Comments and Suggestions for Authors
This Opinion Paper discusses the importance of implementation of newborn screening (NBS) programmes with the help of collaborative support in the light of current global instabilities and recent WHO proposals. The paper is well written and contains important insight to the topic.
Minor issue:
- P3 L111-113: This seems to be meant as a single sentence. Please consider rephrasing, e.g. " The Centers for Disease Control and Prevention (CDC) and the Health Resources and Services Administration (HRSA) provide a shining example of the benefit to national policy provided by a worldwide vision."
• Do you consider the topic original or relevant to the field? Does it address a specific gap in the field? Please also explain why this is/ is not the case.
In my opinion, the topic explaining the importance of international collaborations is relevant to the field. This is especially true for the topic of implementing and maintaining NBS in low-income countries where such programmes are less frequent and need extensive promotion.
• What does it add to the subject area compared with other published material?
The manuscript does not provide an immediate solution; rather, it calls attention to the usefulness of common concepts and sharing existing knowledge. Importantly, this manuscript cites recent proposals of WHO acknowledging NBS as a general tool to decrease morbidity and mortality worldwide.
• What specific improvements should the authors consider regarding the methodology?
In my opinion, the methodology is acceptable as is.
• Are the conclusions consistent with the evidence and arguments presented and do they address the main question posed? Please also explain why this is/is not the case.
Yes, the conclusions are supported by the evidence detailed in the manuscript and address the topic.
• Are the references appropriate?
Yes, the number of references are appropriate but adding 1-3 references on NBS in general could be beneficial.
• Any additional comments on the tables and figures.
There are no tables or figures, and in my opinion, they are not necessary for understanding the scope of this paper.
Author Response
Reviewer 2
We thank Reviewer 2 for supporting newborn screening in this difficult political environment and for the considered review of our submission.
This Opinion Paper discusses the importance of implementation of newborn screening (NBS) programmes with the help of collaborative support in the light of current global instabilities and recent WHO proposals. The paper is well written and contains important insight to the topic.
Minor issue:
- P3 L111-113: This seems to be meant as a single sentence. Please consider rephrasing, e.g. " The Centers for Disease Control and Prevention (CDC) and the Health Resources and Services Administration (HRSA) provide a shining example of the benefit to national policy provided by a worldwide vision."
Thank-you we have replaced the sentences with the improved clarity of the suggested version.
- Do you consider the topic original or relevant to the field? Does it address a specific gap in the field? Please also explain why this is/ is not the case.
In my opinion, the topic explaining the importance of international collaborations is relevant to the field. This is especially true for the topic of implementing and maintaining NBS in low-income countries where such programmes are less frequent and need extensive promotion. Thank-you for your support. - What does it add to the subject area compared with other published material?
The manuscript does not provide an immediate solution; rather, it calls attention to the usefulness of common concepts and sharing existing knowledge. Importantly, this manuscript cites recent proposals of WHO acknowledging NBS as a general tool to decrease morbidity and mortality worldwide. Thank-you for your support.